# Relation between Muscle Force of Knee Extensors and Flexor Muscles and Sleep Quality of Women Soccer Players: A Pilot Study

Ligia Rusu [1,*,†], Aurora Ungureanu Dobre [2,†], Alexandru Gabriel Chivaran [1,†] and Mihnea Ion Marin [3,†]

1   Department of Sport Medicine and Physical Therapy, University of Craiova, 200585 Craiova, Romania
2   Department of Theory and Motric Activity, University of Craiova, 200585 Craiova, Romania
3   Faculty of Mechanics, University of Craiova, 200585 Craiova, Romania
*   Correspondence: ligia.rusu@edu.ucv.ro; Tel.: +40-723867738
†   These authors contributed equally to this work.

**Abstract:** Physical performance in sport performance such as football is influenced by sleep quality, but there are only a few studies about this and the behaviour of muscle force at knee flexors and extensors. The aim of this study is to make an analysis of sleep, an analysis of muscle force parameters (flexors and knee extensors), and a study of the correlation between sleep quality and muscle force. The study includes 18 junior women footballers, average age 14.75 years old, who participated in a training programme of 90 min, three times/week. The evaluation includes sleep quality evaluation using the sleep quality scale (SQS), sleepiness evaluation using the Cleveland Adolescent Sleepiness Questionnaire (CASQ), and muscle force evaluation using dynamometry for knee flexor and extensor muscles. We recorded maximal muscle force, medium maximal muscle force, and duration of maintained muscle force. We observed that more than 83% of subjects scored below 50% on the maximal SQS score and all participants scored less than 40 points (50%) on the CASQ's maxim score, meaning a small level of sleepiness. With regard to muscle force, left–right symmetry for medium maximal force shows a predominance of the right-hand side and the value was 1.64%, for knee extensors and for knee flexors the difference was 7.58%, meaning that for knee extensors the symmetry is better than that of knee flexors. Statistical analysis regarding muscle parameters shows that there are no significant statistical differences between the left- and right-hand sides. Because the duration of the maintained maximal muscle force could indicate muscle fatigue, we observed that there was no link, and that sleep quality or sleepiness do not influence muscle fatigue. Our research concludes that muscle asymmetry of the left–right side of knee extensor and flexor muscles is minimal and does not correlate with sleep quality or sleepiness. There is no statistical correlation between muscle force parameters and sleep quality.

**Keywords:** muscle force; sleep quality; muscle symmetry; athletes





## 1. Introduction

Physical performance in sport performance such as football is influenced by sleep quality and duration. These aspects are the topics of different studies and different ages. At the same time, furthermore, moderate physical exercise is beneficial to sleep quality in both young and old populations [1–3].

Sleep is an important component of health and facilitates the restoration and regeneration of human body function [4]. This aspect has much more importance for athletes because sleep quality influences sport performance [5].

The physiological basis of sleep's role in health is the biologic processes that are developed during the sleep period. These processes include restoring the function of the immune and endocrine systems and also metabolic processes. In the case of athletes, they could have sleep disorders and sometimes have different sleep patterns generated

by different situations that involve the athletes [6]. A good sleep-like quality influences physical performance and this aspect is important in the context of good fitness and limits the risk of injuries. There have been many questionnaires that evaluate the sleep quality of athletes, and the studies refer to sleep disorders, such as the deprivation of sleep, that could affect physical performance. The problem is that there is only a small number of studies on the sleep quality of athletes, or if the studies exist, they include a small number of athletes without underlining the specificity or any link between physical performance, fitness components, and sleep. There are not enough studies on the sleep quality of women football players. In addition, there is a small evidence base about the sleep evaluation. This needs to have evidence-based information because in the present conditions we observe an increase in the number of women footballers, and much more, this implication is more visible in young girls.

Many studies speak about the sleep of women football players and demonstrate that there is a large variability in sleep duration and sleep quality during competitions and that players develop different sleep patterns [4,7,8].

Starting from these aspects, we could consider that there are a lot of different methodology approaches to understanding and interpreting the sleep of athletes and a lot of validated tools for sleep assessment.

Research has been carried out, but studies include a small number of people and are of a short duration. Thus, this topic seems to need a new approach to sport activity. In this context there are some studies that investigate the effect of a decrease in sleep duration on physical performance, such as muscle force [9–12], anaerobic power [13–17], endurance [18], or aerobic capacity.

In their analysis of physical performance, Juliff et al. [19] realised a study about the physiological profile of football players and observed that there are some key points that define this profile, such as aerobic capacity, anaerobic capacity, repeated sprint ability, muscle force, speed, agility and speed reaction, and body composition.

From these, the football player has a speed that is crucial to performance and is correlated with the muscle force of knee flexors and extensors [20]. We know that the quadriceps muscle has an important role in sprinting, jumping, and ball-kicking, and muscle hamstrings contribute to knee flexion and to developing stride power. The problem is regarding muscle symmetry, which influences agility, as Fousekis et al. [21] discuss in their paper. Studies on symmetry show that, in the long term, footballers develop a lot of asymmetry. Maly T. et al. [22], in their study involving women football players, discuss the muscle force of these players' flexors and knee extensors and conclude that they are comparable to the international level in many cases, even though there is not enough information about the thigh muscle behaviour of junior women football players who are involved in national-level competitions. With regard to muscle asymmetry, Maly T. et al. find no differences between dominant and non-dominant lower limbs [22].

With regard to the link between sleep quality and physical performance, the discussion is whether or not a relationship exists, particularly between sleep quality and muscle force.

Sleep is associated with the restoration of mental and tissue regeneration, and sleep deprivation could induce a decrease in cognitive processes, life quality, and physical performance. Furthermore, when sleep quality is poor, anabolic hormones like IGF-1, which play a role in protein synthesis, decrease, and this involves a decrease in muscle mass [23].

The link between sleep deprivation and muscle force is the topic of many studies undertaken with older people, including the relationship between sleep quality, muscle mass, and grip strength in older women [24].

Currently, there is only a small number of studies that examine the effects of bad sleep quality and functional capacity, and also, there are no existing studies on the relationship between sleep and muscle force [25–27].

There are not enough results regarding this aspect, and there are many controversies. Therefore, the aim of our study is to contribute to this aspect and make an analysis of sleep from the quality point of view, as well as an analysis of muscle force (knee flexors and knee

extensors). To do so, we propose to analyse the muscle symmetry (right and left, agonistic and antagonistic muscles) and to study the correlation between sleep quality (through questionnaires) and muscle force.

The novelty of our study is that it uses as its sample junior women football players, whom we consider will enable us to know the real situation, and the results could help trainers in their monitoring the evolution and preparation for high-level competitions. The question is how muscle force could be influenced by sleep quality and whether there is any correlation between sleep quality and knee flexor and extensor muscles. We choose these muscle groups because we consider that these groups are relevant to football players' knee stability during a fast change of movement direction.

## 2. Materials and Methods

Our study includes 18 junior women football players, average age 14.75 years (SD $\pm$ 1.25 yrs), average weight 53.94 kg (SD $\pm$ 10.16 kg), average height 162.63 cm (SD $\pm$ 17.17 cm), and average body mass index (BMI) 20.31 (SD $\pm$ 3.10). The athletes received a code from 1 to 18, and their parents signed the informed consent. The research was undertaken in accordance with the Declaration of Helsinki and was also approved by the ethical committee of the University of Craiova, nr. 15/1.11.2021. The participants are athletes of the *Universitatea Craiova* team. The training programme was 90 min long, 3 times/week and 1 match per week. The training was structured as follows: *Training 1*—develop muscle force, submaximal intensity; *Training 2*—increase the intensity; *Training 3*—technic training at medium intensity.

Structure of the training session: *first part*—prevention and activation (exercises for joints, dynamic and proprioceptive training: 20–25 min). The *technic part* is the second part and includes different systems of activation: touch different portative objects, and the duration is 55–60 min. The last part includes cool-down exercises and consists of easy running, walking, and stretching for 10–15 min.

*Study design*. Evaluations are made regarding sleep quality, sleepiness, and muscle force. The study is a case-series study because the subjects participate at the same training program described before and the proposal is to make a descriptive analysis of sleep quality, sleepiness, and muscle force, but we also try to find an analysis of possible correlation between sleep quality, sleepiness, and muscle force parameters.

### 2.1. Evaluation of Sleep Quality

Sleep quality evaluations are made using the validated sleep quality scale (SQS) [28], which includes 28 items, and uses a grill formed by 4 elements, like the Likert scale, and the participants have to answer how often they have some behaviours regarding sleep (0 = 'rarely', 1 = 'sometimes', 2 = 'often', and 3 = 'almost always'). The scores for items 2 and 5 have to be inversed, according to the scale-fill indications. The total score could be between 0 and 84, and high scores mean acute sleep disorders.

### 2.2. Evaluation of Sleepiness

The sleepiness evaluation was made using the Cleveland Adolescent Sleepiness Questionnaire (CASQ) (acknowledgement for http://www.aasmnet.org/jcsm/ViewAbstract.aspx?pid=26971 on 12 June 2022), which includes 16 items and explains sleepiness during the day. The quantification is between 1 and 5 points, and the maximal score is 80 points.

### 2.3. Evaluation of Muscle Force

We evaluated muscle force using the BioFET equipment dynamometer (MusTec Muscle Dynamic Technology b.v., Louis Christijnstraat 1, 1325 PC Almere, The Netherlands—Figure 1). The BioFET MusTec HD is a handheld dynamometer for evaluating muscle force [29]. It offers a powerful software package that automatically collects data.

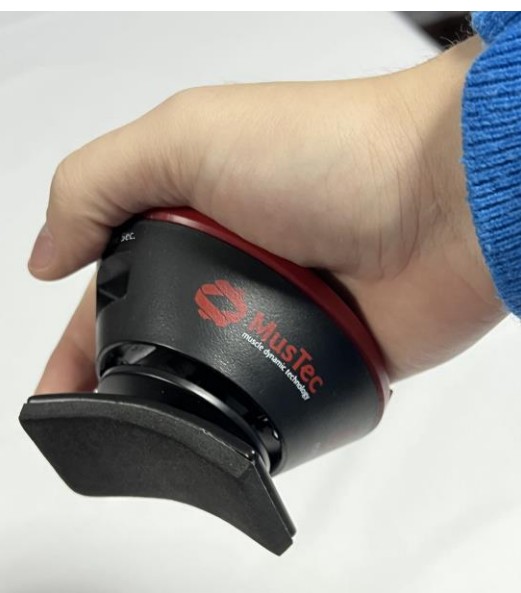

**Figure 1.** BioFET equipment.

This wireless electronic device is extremely handy, ergonomic, and lightweight (250 g), provides accurate records (<1% from registered values), and displays the average and maximum force and time (in seconds).

Forces can be displayed in pounds (lbs), kilos (kg), and newtons (N) in increments of 0.1 units of measurement. Its dimensions are $10 \times 7.5 \times 4.5$ cm (Figure 1), and it can automatically upload measurement data onto a computer via Bluetooth and USB.

We measured the muscle force for flexors and knee extensors and tested the muscle in a supine position for knee extensors and a prone position for knee flexors (see Figures 2 and 3).

For the muscle group we made three measurements for both knees.

The placement of the BioFET is made on the 1/3 distal side of the calf (Figures 2 and 3).

We recorded maximal muscle force, medium maximal muscle force, and duration of maintained muscle force.

Statistical analysis was made using the Microsoft Excel, XLSTAT software package [30].

The statistical analysis included descriptive analysis and a Jarque–Bera test (*JB test*), which gives us information about the normal distribution of parameters.

A *Student's t-test* was applied to reveal any differences between parameter values: maximal muscle force; medium maximal muscle force; duration of maintaining the maximal force for both sides, right and left, but also agonist and antagonist muscles.

The test indicates whether there is a significant difference. We applied the *Student's t-test* for equal means.

For evaluation of the level of correlations between muscle parameters using BioFET and level of sleep quality and sleepiness (based on SQS and CASQ scores), we used Pearson correlation.

To take into account the interactions between 2 variables of sleep and 12 variables of muscles groups, we used MANOVA statistical analysis. This analysis is available in XL STAT software [30]. For comparing the means for each variable, we used the Wilks lambda tests (1932), which is the one of the most important tests, available also in XL STAT software [30].

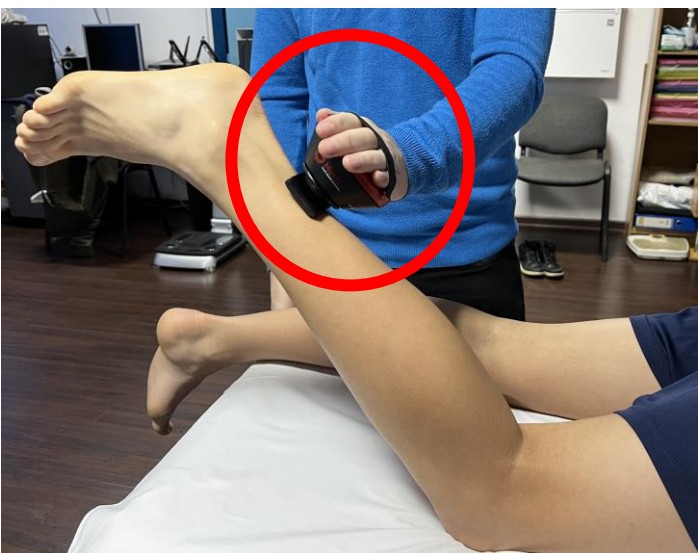

**Figure 2.** Test of knee flexors.

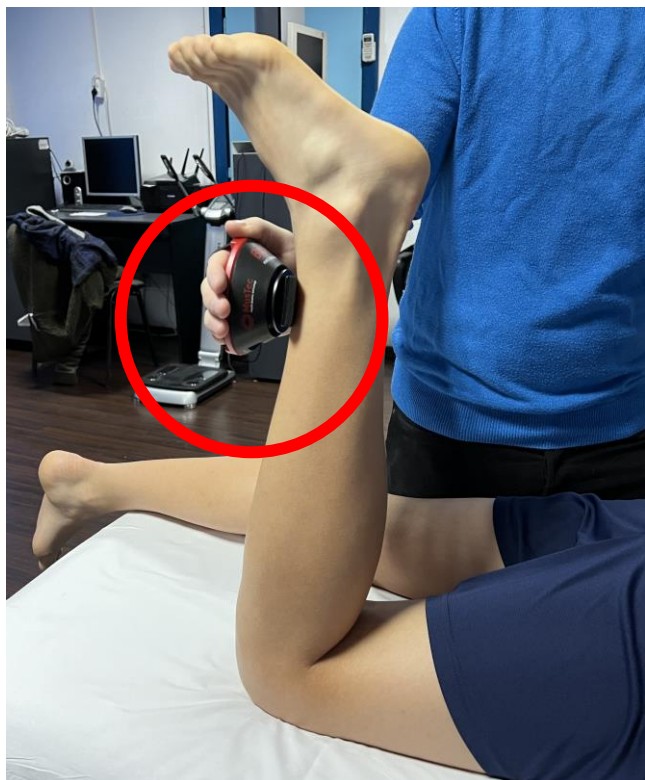

**Figure 3.** Test of knee extensors.

## 3. Results

The SQS scores for all 18 participants are presented in Tables 1 and 2.

**Table 1.** Values for SQS score.

| Subject code | 1 | 2 | 3 | 4 | 5 | 6 | 7 | 8 | 9 | 10 | 11 | 12 | 13 | 14 | 15 | 16 | 17 | 18 |
|---|---|---|---|---|---|---|---|---|---|---|---|---|---|---|---|---|---|---|
| Score SQS | 27 | 46 | 37 | 44 | 51 | 36 | 34 | 29 | 35 | 34 | 39 | 34 | 20 | 24 | 31 | 33 | 20 | 39 |

**Table 2.** Values (extreme, mean, and standard deviation) for SQS.

| Variables | Min. | Max. | Mean | Standard Deviation |
|---|---|---|---|---|
| Scale SQS | 20 | 51 | 34.06 | 8.31 |

Analysis of the SQS score results regarding the percentile repartition of score intervals is presented in Table 3.

**Table 3.** Percentile repartition according to score intervals—SQS.

| Score SQS | Number of Answers | Number of Answers (%) |
|---|---|---|
| 20–30 | 5 | 27.78 |
| 31–35 | 6 | 33.33 |
| 36–40 | 4 | 22.22 |
| 40–45 | 1 | 5.56 |
| >45 | 2 | 11.11 |
| Total | 18 | 100 |

We observed that more than 83% of subjects scored under 50% of the maximal SQS score (84 points).

Ranking the 28 items according to the score, we observed that item 14 has the lowest value (score 0.330), meaning that sometimes sleep did not influence the athletes' appetite (Table 4).

**Table 4.** Ranking of 28 items—SQS.

| Order of Items (1–14) | Item Number | Score | Order of Items (15–28) | Item Number | Score |
|---|---|---|---|---|---|
| 1 | Q 14 | 0.33 | 15 | Q 24 | 1.00 |
| 2 | Q 4 | 0.39 | 16 | Q 11 | 1.11 |
| 3 | Q 7 | 0.39 | 17 | Q 22 | 1.11 |
| 4 | Q 1 | 0.72 | 18 | Q 12 | 1.17 |
| 5 | Q 3 | 0.72 | 19 | Q 28 | 1.18 |
| 6 | Q 9 | 0.72 | 20 | Q 26 | 1.28 |
| 7 | Q 19 | 0.78 | 21 | Q 8 | 1.83 |
| 8 | Q 15 | 0.83 | 22 | Q 13 | 1.94 |
| 9 | Q 17 | 0.83 | 23 | Q 18 | 1.94 |
| 10 | Q 21 | 0.89 | 24 | Q 5 | 2.11 |
| 11 | Q 23 | 0.89 | 25 | Q 27 | 2.11 |
| 12 | Q 25 | 0.89 | 26 | Q 20 | 2.17 |
| 13 | Q 6 | 0.94 | 27 | Q 2 | 2.28 |
| 14 | Q 10 | 1.00 | 28 | Q 16 | 2.56 |

The highest value is for item 16 (score 2.56), which means that all the participants feel good after sleep (Table 4).

Results of the sleepiness questionnaire (CASQ) are presented in Tables 5 and 6.

**Table 5.** Values for CASQ score.

| Subject code | 1 | 2 | 3 | 4 | 5 | 6 | 7 | 8 | 9 | 10 | 11 | 12 | 13 | 14 | 15 | 16 | 17 | 18 |
|---|---|---|---|---|---|---|---|---|---|---|---|---|---|---|---|---|---|---|
| CASQ Score | 28 | 33 | 28 | 34 | 37 | 30 | 34 | 34 | 30 | 29 | 40 | 36 | 26 | 39 | 34 | 37 | 27 | 32 |

**Table 6.** Values (extreme, mean, and standard deviation) for CASQ score.

| Variables | Min. | Max. | Mean | Standard Deviation |
|---|---|---|---|---|
| CASQ score | 26 | 40 | 32.67 | 4.19 |

The results of the percentile repartition of the CASQ score in score intervals are presented in Table 7.

**Table 7.** Percentile repartition of values for CASQ score in score intervals.

| CASQ Score | Number of Answers | Number of Answers (%) |
|---|---|---|
| 20–30 | 7 | 38.89 |
| 31–35 | 6 | 33.33 |
| 36–40 | 5 | 27.78 |
| Total | 18 | 100 |

We observe that all respondents scored below 40 points, which is less than 50% of the CASQ maximum score (80 points). This shows a small level of sleepiness and correlates with the results of the SQS.

The ranking of the items for CASQ shows us that item 16 had the lowest score (value 1.14) and item 8 had the highest (value 3.67). This means that sometimes the sleepiness level could be great without affecting the after-school activity programme (Table 8).

**Table 8.** Ranking of items on CASQ, based on the answers.

| Order of Items (1–8) | Item number | Score | Order of Items (9–16) | Item Number | Score |
|---|---|---|---|---|---|
| 1 | Q 16 | 1.14 | 9 | Q 9 | 2.24 |
| 2 | Q 3 | 1.24 | 10 | Q 5 | 2.38 |
| 3 | Q 10 | 1.29 | 11 | Q 11 | 2.43 |
| 4 | Q 15 | 1.29 | 12 | Q 7 | 2.67 |
| 5 | Q 6 | 1.33 | 13 | Q 12 | 2.76 |
| 6 | Q 1 | 1.43 | 14 | Q 2 | 3.05 |
| 7 | Q 4 | 1.52 | 15 | Q 13 | 3.24 |
| 8 | Q 14 | 2.05 | 16 | Q 8 | 3.67 |

The results of the muscle force evaluation (knee flexors and knee extensors) are presented in Table 9, showing the minimum, maximum, and mean values for each parameter.

**Table 9.** Statistical description of muscle force parameters.

| Variables | Min. | Max. | Mean | Standard Deviation |
|---|---|---|---|---|
| EGS Medium of maximal force (N) | 87.44 | 190.74 | 136.85 | 31.36 |
| EGS Maximal force (N) | 107.87 | 238.30 | 165.65 | 36.08 |
| EGS Duration of maintained maximal force (s) | 0.71 | 1.97 | 1.22 | 0.36 |
| EGD Medium of maximal force (N) | 80.74 | 201.20 | 138.54 | 36.98 |
| EGD Maximal force (N) | 111.31 | 215.75 | 159.30 | 32.70 |
| EGD Duration of maintained maximal force (s) | 0.49 | 2.14 | 1.15 | 0.46 |
| FGS Medium of maximal force (N) | 63.74 | 151.68 | 112.86 | 19.23 |
| FGS Maximal force (N) | 112.78 | 168.67 | 135.42 | 14.20 |
| FGS Duration of maintained the maximal force (s) | 0.59 | 2.20 | 1.29 | 0.47 |
| FGD Medium of maximal force (N) | 59.33 | 160.50 | 118.64 | 27.74 |
| FGD Maximal force (N) | 108.85 | 173.09 | 138.30 | 20.11 |
| FGD Duration of maintained maximal force (s) | 0.52 | 2.07 | 1.09 | 0.38 |

Legend: EGS = left knee extensors; FGS = left knee flexors; EGD = right knee extensors; FGD = right knee flexors.

Left/right symmetry for medium maximal force demonstrates a predominance of the right-hand side, and the value is 1.64% for knee extensors. For knee flexors, the same parameter has a dominance on the right-hand side and a difference of 7.58%. So, we can conclude that for knee extensors the symmetry is better for knee flexors in terms of medium maximal force.

With regard to maximal force, left and right symmetry shows a dominance of the left-hand side and a difference of 3.12% for knee extensors. For knee flexors the same parameter has a dominance on the right-hand side, and symmetry is 2.25%.

In this case, the symmetry is close between knee extensors and knee flexors.

Left and right symmetry for duration of maintained maximal force shows a dominance of the left hand, and the value is 2.02% for knee extensors. For knee flexors, the same parameter has a dominance for the left-hand side, and symmetry is 4.29% (see Table 10).

**Table 10.** Left and right symmetry (%).

| Subject Code | EG Medium of Maximal Force Difference Left/Right (%) | EG Maximal Force Difference Left/Right (%) | EG Duration of Maintained Maximal Force Difference Left/Right (%) | FG Medium of Maximal Difference Left/Right (%) | FG Maximal Force Difference Left/Right (%) | FG Duration of Maintained Maximal Force Difference Left/Right (%) |
|---|---|---|---|---|---|---|
| 1 | 6.25 | 2.87 | 27.92 | −44.74 | −16.49 | 12.50 |
| 2 | −22.12 | −2.09 | 39.31 | −17.18 | −2.71 | 0.69 |
| 3 | −3.30 | 1.45 | 18.75 | −22.39 | −17.04 | 34.18 |
| 4 | −11.52 | −2.72 | 5.88 | −83.61 | −0.39 | −17.43 |
| 5 | −9.92 | −5.46 | −77.78 | 10.62 | −6.95 | −57.63 |
| 6 | −2.17 | 6.74 | 46.03 | −3.34 | −2.62 | 35.54 |
| 7 | 0.51 | 9.91 | −33.80 | 50.34 | 15.98 | 40.23 |
| 8 | 0.59 | 12.46 | 26.11 | −7.24 | −9.76 | −165.38 |
| 9 | −9.38 | −0.24 | 25.00 | −10.56 | 5.15 | 24.14 |
| 10 | 15.71 | −8.73 | 45.30 | 4.94 | 13.04 | −50.00 |
| 11 | −11.56 | 19.14 | 2.04 | −28.79 | −22.18 | 39.11 |
| 12 | 14.01 | 13.59 | −26.67 | 15.88 | 7.61 | −3.64 |
| 13 | −9.51 | 0.30 | −36.00 | 20.23 | 10.94 | 11.64 |
| 14 | −11.93 | −8.19 | 14.49 | 27.52 | −0.38 | 41.54 |
| 15 | −12.00 | −17.82 | 13.66 | −4.07 | 2.60 | 22.22 |
| 16 | −15.70 | 9.85 | −32.92 | 3.97 | 11.55 | 45.00 |
| 17 | 44.93 | 20.34 | 56.64 | −29.28 | −13.69 | 74.54 |
| 18 | 7.60 | 4.67 | −77.57 | −18.74 | −15.13 | −10.00 |
| **Average value** | **−1.64** | **3.12** | **2.02** | **−7.58** | **−2.25** | **4.29** |

With regard to this parameter, we see a left/right symmetry that is also close between knee extensors and knee flexors.

A statistical analysis of muscle parameters was made using the *Student t-test* ($p = 0.06$) and shows that there are no significant statistical differences between the left- and right-hand sides, as shown in Table 11.

**Table 11.** Statistical results, left- and right-hand sides.

| Parameters | Medium Maximal Force EGS-EGD | Maximal FORCE EGS-EGD | Duration of Maintained Maximal Force EGS-EGD | Medium Maximal Force FGS-FGD | Maximal Force FGS-FGD | Duration of Maintained Maximal Force FGS-FGD |
|---|---|---|---|---|---|---|
| Difference | −1.687 | 6.346 | 0.075 | −5.778 | −2.886 | 0.201 |
| *p*-value (Two-tailed) | 0.750 | 0.137 | 0.501 | 0.396 | 0.443 | 0.191 |

About agonist and antagonist muscle symmetry we observe that there is a significant statistical difference regarding muscle force, but there is no significant statistical difference regarding duration of maintained maximal muscle force, as can be seen in Table 12.

**Table 12.** Statistical results for agonist/antagonist muscles.

| Parameters | Medium Maximal Force EGS-FGD | Maximal Force EGS-FGD | Duration of Maintained Maximal Force EGS-FGD | Medium Maximal Force EGS-FGD | Maximal Force EGS-FGD | Duration of Maintained Maximal Force EGS-FGD |
|---|---|---|---|---|---|---|
| Difference | 23.989 | 30.234 | −0.067 | 19.898 | 21.002 | 0.058 |
| *p*-value (Two-tailed) | **0.006** | **0.001** | 0.597 | **0.016** | **0.001** | 0.619 |

Because the duration of maintained maximal muscle force is one of the elements that could indicate muscle fatigue and also the decrease in muscle performance, we consider that is important to analyse whether or not there is a correlation between the average values of SQS, CASQ scores, and the average duration of maximal muscle force for knee extensors and flexors, left and right sides.

The results are presented in Tables 13 and 14.

**Table 13.** Correspondence between each SQS score group and average duration of maintained muscle force (knee flexors and knee extensors).

| Score SQS (Intervals) | EGS Average Duration of Maintained Maximal Muscle Force (s) | EGD Average Duration of Maintained Maximal Muscle Force (s) | FGS Average Duration of Maintained Maximal Muscle Force (s) | FGD Average Duration of Maintained Maximal Muscle Force (s) |
|---|---|---|---|---|
| 20–30 | **1.41** | 1.09 | **1.53** | 1.23 |
| 31–35 | 1.19 | 1.21 | 1.26 | 1.03 |
| 36–40 | 1.07 | 1.08 | 1.22 | 0.90 |
| 40–45 | 1.19 | 1.12 | 1.09 | **1.28** |
| >45 | 1.18 | **1.24** | 1.02 | 1.18 |

**Table 14.** Correspondence between each CASQ score group and average duration of maintain the muscle force (knee flexors and knee extensors).

| CASQ Score (Intervals) | EGS Average Duration of Maintained Maximal Muscle Force (s) | EGD Average Duration of Maintained Maximal Muscle Force (s) | FGS Average Duration of Maintained Maximal Muscle Force (s) | FGD Average Duration of Maintained Maximal Muscle Force (s) |
|---|---|---|---|---|
| 20–30 | 1.15 | 0.81 | 1.29 | 0.94 |
| 31–35 | **1.31** | 1.26 | 1.09 | **1.25** |
| 36–40 | 1.21 | **1.48** | **1.53** | 1.10 |

An analysis of the results shows us that the lowest SQS score is correlated with an increase in the average duration of maintained maximal muscle force, in the case of knee extensors and flexors. With regard to the relationship between this parameter and the sleep scales (SQS and CASQ) score evaluation, we can see that there is no relationship and that sleep quality and/or sleepiness do not influence muscle fatigue.

In the same context, we also made a Pearson correlation between the parameters (Tables 15–17).

In Tables 18–20 we present the p values for Pearson coefficients that are presented in Tables 15–17.

**Table 15.** Pearson correlation between muscle parameters of left and right knee extensors.

| Variables | EGS | EGS | EGS | EGD | EGD | EGD |
|---|---|---|---|---|---|---|
| EGS Medium maximal muscle force (N) | 1 | **0.841** | 0.086 | **0.803** | **0.870** | −0.240 |
| EGS Maximal muscle force (N) | | 1 | 0.051 | **0.757** | **0.879** | −0.230 |
| EGS Duration of maintained maximal muscle force (s) | | | 1 | 0.078 | 0.121 | 0.374 |
| EGD Medium maximal muscle force (N) | | | | 1 | **0.925** | −0.114 |
| EGD Maximal muscle force (N) | | | | | 1 | −0.214 |
| EGD Duration of maintained maximal muscle force(s) | | | | | | 1 |

The bold values are different from 0 with a significance level $\alpha$ = 0.05.

**Table 16.** Pearson correlation between muscle parameters of left and right knee extensors and flexors.

| Variables | FGS | FGS | FGS | FGD | FGD | FGD |
|---|---|---|---|---|---|---|
| EGS Medium maximal muscle force (N) | 0.231 | **0.644** | 0.018 | **0.727** | **0.823** | −0.149 |
| EGS Maximal muscle force (N) | 0.244 | **0.517** | 0.259 | **0.758** | **0.811** | −0.046 |
| EGS Duration of maintained maximal muscle force (s) | −0.129 | −0.049 | 0.200 | 0.228 | 0.127 | **0.501** |
| EGD Medium maximal muscle force (N) | 0.170 | **0.538** | −0.061 | **0.556** | **0.611** | 0.025 |
| EGD Maximal muscle force (N) | 0.226 | **0.583** | 0.115 | **0.713** | **0.741** | 0.006 |
| EGD Duration of maintained maximal muscle force | −0.140 | −0.389 | 0.030 | −0.177 | −0.266 | 0.324 |
| FGS Medium maximal muscle force (N) | 1 | **0.675** | −0.140 | 0.324 | 0.400 | −0.073 |
| FGS Maximal muscle force (N) | | 1 | −0.068 | **0.582** | **0.635** | −0.157 |
| FGS Duration of maintained maximal muscle force (s) | | | 1 | 0.007 | −0.007 | −0.079 |
| FGD Medium maximal muscle force (N) | | | | 1 | **0.901** | 0.076 |
| FGD Maximal muscle force (N) | | | | | 1 | −0.156 |
| FGD Duration of maintained maximal muscle force (s) | | | | | | 1 |

The bold values are different from 0 with a significance level $\alpha$ = 0.05.

**Table 17.** Pearson correlation between SQS, CASQ scores and muscle parameters of left and right knee extensors and flexors.

| Variables | SQS Score | CASQ Score |
|---|---|---|
| EGS Medium maximal muscle force (N) | −0.057 | **−0.475** |
| EGS Maximal muscle force (N) | 0.022 | −0.163 |
| EGS Duration of maintained maximal muscle force (s) | −0.188 | 0.106 |
| EGD Medium maximal muscle force (N) | 0.224 | −0.167 |
| EGD Maximal muscle force (N) | 0.099 | −0.233 |
| EGD Duration of maintained maximal muscle force(s) | 0.165 | **0.478** |
| FGS Medium maximal muscle force (N) | 0.041 | 0.022 |
| FGS Maximal muscle force (N) | 0.051 | −0.346 |
| FGS Duration of maintained maximal muscle force (s) | **−0.478** | 0.145 |
| FGD Medium maximal muscle force (N) | 0.237 | −0.215 |
| FGD Maximal muscle force (N) | 0.124 | −0.256 |
| FGD Duration of maintained maximal muscle force (s) | 0.013 | 0.199 |
| SQS Score | 1 | 0.388 |
| CASQ Score | | 1 |

The bold values are different from 0 with a significance level $\alpha$ = 0.05.

**Table 18.** *p*-values for Pearson correlation between muscle parameters of left and right knee extensors.

| Variables | EGS | EGS | EGS | EGD | EGD | EGD |
|---|---|---|---|---|---|---|
| EGS Medium maximal muscle force (N) | 0 | **<0.0001** | 0.733 | **<0.0001** | **<0.0001** | 0.338 |
| EGS Maximal muscle force | | 0 | 0.842 | **0.000** | **<0.0001** | 0.358 |
| EGS Duration of maintained maximal muscle force (s) | | | 0 | 0.759 | 0.631 | 0.127 |
| EGD Medium maximal muscle force (N) | | | | 0 | **<0.0001** | 0.654 |
| EGD Maximal muscle force | | | | | 0 | 0.393 |
| EGD Duration of maintained maximal muscle force (s) | | | | | | 0 |

**Table 19.** *p*-values for Pearson correlation between muscle parameters of left and right knee extensors and flexors.

| Variables | FGS | FGS | FGS | FGD | FGD | FGD |
|---|---|---|---|---|---|---|
| EGS Medium maximal muscle force (N) | 0.356 | **0.004** | 0.942 | **0.001** | **<0.0001** | 0.555 |
| EGS Maximal muscle force | 0.330 | **0.028** | 0.300 | **0.000** | **<0.0001** | 0.856 |
| EGS Duration of maintained maximal muscle force (s) | 0.609 | 0.847 | 0.425 | 0.363 | 0.614 | **0.034** |
| EGD Medium maximal muscle force (N) | 0.500 | **0.021** | 0.810 | **0.017** | **0.007** | 0.920 |
| EGD Maximal muscle force | 0.366 | **0.011** | 0.648 | **0.001** | **0.000** | 0.982 |
| EGD Duration of maintained maximal muscle force (s) | 0.580 | 0.111 | 0.907 | 0.483 | 0.286 | 0.190 |
| FGS Medium maximal muscle force (N) | 0 | **0.002** | 0.580 | 0.189 | 0.100 | 0.774 |
| FGS Maximal muscle force | | 0 | 0.789 | **0.011** | **0.005** | 0.533 |
| FGS Duration of maintained maximal muscle force (s) | | | 0 | 0.979 | 0.978 | 0.757 |
| FGD Medium maximal muscle force (N) | | | | 0 | **<0.0001** | 0.765 |
| FGD Maximal muscle force | | | | | 0 | 0.536 |
| FGD Duration of maintained maximal muscle force (s) | | | | | | 0 |

**Table 20.** *p*-values for Pearson correlation between SQS, CASQ scores and muscle parameters of left and right knee extensors and flexors.

| Variables | SQS Score | CASQ Score |
|---|---|---|
| EGS Medium maximal muscle force (N) | 0.821 | **0.046** |
| EGS Maximal muscle force (N) | 0.932 | 0.519 |
| EGS Duration of maintained maximal muscle force (s) | 0.455 | 0.677 |
| EGD Medium maximal muscle force (N) | 0.372 | 0.507 |
| EGD Maximal muscle force (N) | 0.696 | 0.351 |
| EGD Duration of maintained maximal muscle force | 0.514 | **0.045** |
| FGS Medium maximal muscle force (N) | 0.873 | 0.931 |
| FGS Maximal muscle force (N) | 0.840 | 0.160 |
| FGS Duration of maintained maximal muscle force (s) | **0.045** | 0.566 |
| FGD Medium maximal muscle force (N) | 0.345 | 0.392 |
| FGD Maximal muscle force (N) | 0.625 | 0.306 |
| FGD Duration of maintained maximal muscle force (s) | 0.960 | 0.429 |
| SQS Score | 0 | 0.112 |
| CASQ Score | | 0 |

We observe in Table 17 that the Pearson coefficients are less than 0.5, which means that there is weak correlation between muscle parameters on one side and the SQS and CASQ score on the other side.

In Table 20 we observed that p values associated with correlation coefficients presented in Table 17 are below the threshold 0.05 only for good correlations. For these correlations, the Pearson coefficient is greater than the value 0.47. The good correlations are between the following pairs of variables: EGS medium maximal muscle force-CASQ, EGD duration of maintained maximal muscle force-CASQ, and FGS duration of maintained maximal muscle force-SQS.

For all other correlations between sleep parameters and muscle parameters, we have weak correlation (Pearson coefficient is less than 0.4) and p value is more than 0.05.

These data allow us to affirm, with a probability above 95%, that there are no moderate or good correlations between sleep and muscle parameters.

For calculations of the correlation power for this small number of subjects, we use the software *GPower version 3.1.9.6* (https://g-power.apponic.com/ accessed on 12 June 2022), with post hoc power analysis. The parameters used were the Pearson correlation coefficient = 0.478, the sample size = 18 subjects, and the significance level $\alpha$ = 0.05. Using this software and these data we determine that the power test for Pearson's correlation is 0.69.

Multivariate analysis was made using multivariate tests, the MANOVA (multivariate analysis of variance) test, from XL STAT software.

We made four types of tests:

- Explanatory variables—we used the results of sleep questionnaires (SQS and CASQ);
- Dependent variables—we used the results on duration of maintained maximal force for all muscle groups studied by us and the medium of maximal force for all muscle groups.

The results of questionnaires were divided into three intervals, as shown in Table 21.

**Table 21.** Intervals groups for CASQ and SQS scores.

| Interval Groups | CASQ Scores | SQS Score |
| --- | --- | --- |
| 1 | 27–29 | 20–29 |
| 2 | 30–35 | 30–39 |
| 3 | 35–40 | 40–50 |

The most frequently used test is the Wilks lambda test. **Using the MANOVA test we make the analysis, and we obtained the values of Wilks lambda parameters test** (lambda, *p*, F observed, and critical values), which are presented in Table 22.

**Table 22.** Parameter values of the Wilks lambda test.

| Dependent Variables | Explanatory Variables | Lambda | F Observed Values | F Critical Value | *p*-Value |
| --- | --- | --- | --- | --- | --- |
| Duration of maintained maximal force (s) | Sleepiness | 0.548 | 1.052 | 2.355 | 0.427 |
| Medium of maximal force (N) | Sleepiness | 0.508 | 1.208 | 2.355 | 0.336 |
| Duration of maintained maximal force (s) | Sleep Quality | 0.686 | 0.621 | 2.355 | 0.752 |
| Medium of maximal force (N) | Sleep Quality | 0.451 | 1.467 | 2.355 | 0.221 |

The null hypothesis **Wilks lambda test** is that the explanatory variables have no significant effect on the dependent variables.

As we can see in Table 22, *p* values are at a more-than significant level ($\alpha = 0.05$) for all tests, so the null hypothesis cannot be rejected.

The conclusion, based on the multivariate tests, shows that all aspects regarding sleep cannot be use like predictors of duration of maintained maximal force and medium maximal force. This is according to the correlation analysis (see Table 17).

## 4. Discussion

Our study demonstrates that, even if we have a small number of subjects, a complex approach of how sleep quality and daily sleepiness can affect muscle force is needed to determine the possibilities of how trainers can choose restoration tools in the post-effort period and how to sustain the athlete's efforts.

*About the sleep quality*, we observed no specific pattern of sleep quality, but the small values of scores indicate a good sleep quality.

*With regard to muscle force*, we observe an asymmetry of left- and right-hand sides for many of the muscle parameters, but in terms of the relationship between agonist and antagonist muscles, we observe a significant statistical difference for knee extensors and knee flexors. This is in accord with the results of Maly T. et al. [20], which show that an important asymmetry also exists between knee extensors and knee flexors at high angle speed. They [20] observe that less than 50% of women football players have left and right muscle asymmetry, with a dominance of knee flexors. This aspect is also presented in our research. In the long term, we consider that it is important to monitor this correlation and maybe conduct an in-depth study that also includes other factors. This asymmetry could generate maladaptive situations and increase the risk of injuries.

*Relationship between physical activity and sleep quality*: as we can observe, in our pilot study the physical effort during training does not negatively influence sleep quality, and

sleepiness is reduced, and this allows women football players to continue their activities, even at the end of the school programme and training. This approach to physical activity in connection with a training programme is an innovative approach compared with other studies. This is because many other studies are focused on the relationship between sleep deprivation and the status of physical activity.

With regard to physical performance and sleep quality, as stated before, we find no significant relationship and this is in accord with the results of other authors [31]. They demonstrate that younger athletes who sleep less than 8 h a night have a high risk of injuries. The same authors underline that even if athletes slept more and their physical performance increases, they found no relationship between sleep quality and performance results, and the conclusions are not yet clear [31]. In line with the previous studies regarding physical performance and sleep duration, we underline that our study demonstrates that there is no link between sleep quality and physical performance, and this aspect represents a new approach to this relationship and the innovative characteristic of our study. This is because much more of this aspect is still in controversy. In this context, we consider that our study is innovative because it tries to analyse the relationship between muscle force (one component of physical performance and fitness) and sleep quality. Many other studies are focused on sleep duration but not on the quality of sleep. Our study results are in contradiction with those of Bromley et al.'s study [31], which relates to people who have a low level of sleep quality, as well as sleepiness and a reduced physical activity level.

In the same context, as we analysed *regarding muscle parameters and sleep quality*, in their study, Knowles et al. reviewed how resistance exercises could be influenced by sleep quality, in terms of the effects of sleep deprivation and sleep restriction. The result is that sleep deprivation has a small effect under muscle force during resistance exercises, but sleep restrictions could lead to reducing the muscle force in the kinetic muscle chain [32].

The problem with sleep quality and muscle force is also approached by Chen et al. [33,34] and they suggest that even if sleep is a part of the homeostasis process and important for physical performance, short sleep duration is associated with an increase in the inflammatory process in younger people. This aspect is considered to be an important factor that affects muscle force.

The conclusion of their study is that the relationship between sleep quality and muscle force in younger people is unknown. This study involved the relationship between sleep quality and hand-grip strength. Our study contributes to and completes the previous study because we analyse the relationship between thigh muscle force and sleep quality and demonstrate results that are presented using statistical analysis.

With regard to sleep quality and muscle force, there is much controversy, but many of the studies maintain that sleep deprivation does not affect muscle force. We speak about right- and left-hand grips or the leg and back musculature, and this aspect about the leg is also present in our research.

In a similar vein, Rasheed et al. [35] study the physical activity, postural stability, and muscle force of two younger groups of athletes. One group has normal sleep and one low sleep quality. The authors observe that there are significant differences between the two groups and this aspect is also presented by Bambaeichi et al. [10], who followed the same relationship in a group of younger people who sleep only 2.5 h by night.

Even if there are a lot of studies regarding the effect of sleep deprivation, we observe that the results are contradictory, as was also observed by Fullagar et al. [36]. With regard to muscle force, the effect of sleep deprivation does not influence muscle contraction and aerobic capacity effort. In addition, according to Knowles [32] and as we noted before, sleep restriction generates the function of the whole kinetic chain of lower limbs, but not only one joint. This agrees with our observation and also concurs with Kujawa et al. [37], who measured the effect of 24 hours' sleep deprivation on the knee extensors and knee flexors, and the results show that the effects are different, which means that a decrease in the knee extensors' muscle force does not influence the muscle force of the knee flexors.

Our approach regarding the behaviour of the knee muscles is also supported by Kim et al., who demonstrate that muscle force of the lower limbs provided information about fitness [38,39], but they also observe that hand-grip strength is important for measuring the global muscle force and is correlated with knee extensors' muscle force. With regard to this study, we can say that our study is novel because our measurements are made for knee flexors and extensors muscle force in relation to sleep quality.

Our results could be considered also in the clinical approach regarding how the different treatment could change the muscle proprieties because in many clinical studies life-quality assessment also includes sleep-quality evaluation. For example, in upper neuron syndrome, life quality decreases and also the performance of daily living. By this way, muscle force and muscle tone suffer, and the main tool is to use different physical therapy interventions but also specific treatments, such as radial extracorporeal shockwave therapy, which reduce the spasticity [40]. These results lead to increases in the level of physical activity participation, to increases in the level of participation in daily living activities, and of course, to having an improvement of life quality, including also sleep quality.

## 5. Conclusions

Our study concludes that there is a need to extend the research to a large number of people and to optimise the methods for sleep evaluation. Our research concludes that muscle asymmetry of the left–right sides of knee extensors and knee flexors muscles is minimal and does not correlate with sleep quality or sleepiness. There is no statistical correlation between muscle force parameters (medium maximal muscle force, maximal muscle force, and duration of maintained maximal muscle force) and sleep quality.

The mechanisms that could influence sleep quality under the muscle force we can find in the adaptive mechanisms of muscles in different physical activities, if we speak about the athletes. According to our results, we can conclude that the relationship between sleep quality and muscle force is unclear, yet we do not know exactly what the mechanism is that could be behind the muscle changes associated with lower sleep quality.

A complex approach to how sleep quality and/or daily sleepiness can affect muscle force could help trainers in developing a programme for restoring functional capacity, post-effort.

Future research is needed to analyse the correlation between sleep quality, sleep duration, and muscle force but also the correlation with intervention for different training and how the athletes follow the methods to restore their functional capacity and fitness level.

**Limitations of the study**: a small number of subjects means the analysis of a small number of muscles. Our study concludes that there is a need to extend the research to a large number of people and to optimise the methods for sleep evaluation. The completion of the questionnaires is subject to the subjectivity of the people who fill them in. Some answers could be influenced by other conditions and by the respondents' disposition. In addition, the results of the dynamometry could have errors generated by the possibility of muscle fatigue being present without clinical expression.

**Author Contributions:** L.R.—conceptualisation, methodology, writing—draft preparation, investigation; A.U.D.—methodology, formal analysis, resources; A.G.C.—formal analysis, resources; M.I.M.—software, data curation, supervision. All authors have read and agreed to the published version of the manuscript.

**Funding:** This research received no external funding.

**Institutional Review Board Statement:** Ethical approval-15/1/11/2021. The study was conducted in accordance with the Declaration of Helsinki and approved by the Ethics Committee of the University of Craiova, Faculty of Physical Education and Sport—Sport Medicine Department.

**Informed Consent Statement:** Informed consent was obtained from all subjects involved in the study. Written informed consent has been obtained from the patient(s) to publish this paper.

**Data Availability Statement:** Not applicable.

**Conflicts of Interest:** The authors declare no conflict of interest.

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
