# Peer review of "Relation between Muscle Force of Knee Extensors and Flexor Muscles and Sleep Quality of Women Soccer Players: A Pilot Study"

_applsci, doi:10.3390/app13042289_

Round 1
Reviewer 1 Report
The study is well organized and written well. Just a little bit changes/revisions are required. See my specific comments below;
1. Revise the abstract more precisely and comprehensively
2. Add the study variables correlation matrix table (Results Section)
3. Revise your conclusion. In its current form, it doesn't reflect the main conclusion theme according to your research objectives
4. Elaborate on study limitations in detail
5. What are the implications and future scope of the study? Add it after the conclusion
Author Response
Thank you very much for time spent and sugestion for improve the paper quality.

Reviewer 2 Report
The Authors of the manuscript decided to analyze the relationship between sleep quality and muscle strength in soccer players. This is very valuable approach; however I have numerous doubts about this study.
First of all, the manuscript is full of grammar and punctuation errors that altogether make it hard to follow the information that Authors would like to present.
Then, the study group is extremely small, please provide relevant calculations for the choice of this number.
Line 64 – the Authors claim that there are not any validated tools for sleep quality assessment available, while there some e.g. PSQI.
Please underline more, why this study is innovative comparing to other ones, already conducted upon this topic.
Line 145 – please provide details of equipment producer (location) according to scientific article standards.
The figure present that the measuring device is hold by the operator, who can also add their own pressure to the measured strength. How was this aspect solved?
Please provide the full questionnaire as supplementary material.
The results section contains interpretation and conclusions that should be presented in discussion section.
There are parts of the article template that have not been deleted before submission, please correct it and kindly pay more attention to overall preparation of the manuscript.
I encourage the Authors to enroll more participants into the study and submit more refined manuscript.
Author Response
Dear reviewer,
Thank you very much for time spent and for your opinions about our work. All help us to improve the paper quality.

Reviewer 3 Report
Dear Authors,
Your paper seems interesting, but some aspects have to be addressed.
Line 16: there is wrongly repeated two times.
Line 19: and to study
Line 22: the sleep of sleepiness…this concept is not clear, please reformulate
Line 151: from 1 to 18.
Study design is not clear. Is this a case series study? Please explain this aspect.
The discussion seems still poor and in some parts confusing. I suggest to reorganize it, and to briefly integrate it with a consideration about the muscles properties and the capacity of treatments to change them. To do that, I suggest the following reference:
Megna M, Marvulli R, Farì G, Gallo G, Dicuonzo F, Fiore P, Ianieri G. Pain and Muscles Properties Modifications After Botulinum Toxin Type A (BTX-A) and Radial Extracorporeal Shock Wave (rESWT) Combined Treatment. Endocr Metab Immune Disord Drug Targets. 2019;19(8):1127-1133. doi: 10.2174/1871530319666190306101322. PMID: 30843498.
Best regards
Author Response
Dear reviewer,
Thank you very much for your sugesstion.

Round 2
Reviewer 2 Report
Dear Authors,
Thank you very much for your corrections. Unfortunately, there are still lots of grammar and spelling errors that make the manuscript hard to understand. I suggest using an English language service or native speaker consultation.
Line 39 – The Authors mention “studies” (plural) and refer to only one. Please provide more relevant citations.
Tables 5, 6, 11 and 12 – this data can be presented as a regular text. It does not require a separate table.
As the Authors analyzed the quality of sleep, in opposition to sleep duration, I suggest performing a multivariate analysis regarding the more predictors. The correlations alone are a too simple form of statistical analysis.
Author Response
Than you very much for your support and suggest.

Round 3
Reviewer 2 Report
The extensive English revision increased the quality of the manuscript presentation. Thank you very much for these broad corrections.
I have one major comment essential for the result interpretation and presentation:
In the result section you provide only the R-value and interpret them in terms of significance, while the is no information about the p-value. The R-value provides information about the strength of correlation, but the p-value gives information about the significance! On what basis do you form your conclusions?
Table 15a and 15b, there is only information about R-value (? – it is not precised), while there is not any information about the p-value for this analysis.
Line 332-334 – R-value lower than 0.5 means weak/moderate correlation. The significance is dependent on the p-value which is not provided.
I still have also few minor comments:
Line 76-79 – The Authors mention „many studies”. Please kindly provide relevant references.
Line 126 - The Authors mention „many studies” and cite only one study. Please kindly provide relevant references.
Line 181 – I suggest acknowledging the source link in the relevant reference.
Line 187 – “Evaluated” – I suggest using the past tense instead of the present tense.
Line 226 – There are results of Pearson's correlation, while there is not any information about it in the methods section. Please correct.
Line 226 – Please describe the statistical methods used in the methods section (e.g. information from lines 340-342, 368-380)
Line 235 – “observed” – I suggest using the past tense instead of the present tense.
Table 13 and 14 – The values presented in te table are the means for each subgroup, not correlations like the title claims. Please correct it. There are not provided any R and p values to analyze correlations.
Line 530-531 – There are still lines from the template. Please correct it.
Author Response
Title: Relation between muscle force of knee extensors and flexors muscles and sleep quality at women soccer players: A pilot study
Dear reviewer
Thank you again for your time spent for evaluate our paper. I hope that now our work will be appreciate and hope that our answers are good.
Thank you again for all.
The extensive English revision increased the quality of the manuscript presentation. Thank you very much for these broad corrections.
I have one major comment essential for the result interpretation and presentation:
In the result section you provide only the R-value and interpret them in terms of significance, while the is no information about the p-value. The R-value provides information about the strength of correlation, but the p-value gives information about the significance! On what basis do you form your conclusions?
Table 15a and 15b, there is only information about R-value (? – it is not precised), while there is not any information about the p-value for this analysis.
Answer: In Table 15a and 15c we are presenting the Pearson correlation. The bold values are different from 0 with a significance level alpha=0.05.
In Tables 15d-15f we presented the p values for Pearson coefficients that are presented in Tables 15a-15c.
Line 332-334 – R-value lower than 0.5 means weak/moderate correlation. The significance is dependent on the p-value which is not provided.
Answer: We observed, in Table 15c that Pearson coefficients are less than 0.5, which means that there is weak correlation between muscle parameters on one side and the SQS, CASQ score on the other side. In Table 15f we observed that p values associated with correlation coefficients presented in Table 15c , are less then 0.05 only for good correlations that exist for sleep and muscle parameters () (EGS Medium maximal muscle force-CASQ, EGD Duration of maintained maximal muscle force-CASQ and FGS Duration of maintained maximal muscle force-SQS )(for these Pearson coefficient is more then 0.47).
For all other correlations between sleep parameters and muscle parameters, we have weak correlation ( Pearson coefficient is less then 0.4) but p value is more then 0.05.
These results allow to say that the a probability is 95% and confirm that between sleep parameters and muscle parameters are not moderate or good correlations.
I still have also few minor comments:
Line 76-79 – The Authors mention „many studies”. Please kindly provide relevant references.
Answer: DONE
Line 126 - The Authors mention „many studies” and cite only one study. Please kindly provide relevant references.
Answer: DONE
Line 181 – I suggest acknowledging the source link in the relevant reference.
Answer: DONE
Line 187 – “Evaluated” – I suggest using the past tense instead of the present tense.
Answer: DONE
Line 226 – There are results of Pearson's correlation, while there is not any information about it in the methods section. Please correct.
Answer: For evaluation the level of correlations between muscle parameters using BioFet and level of sleep quality and sleepiness ( based on SQS and CASQ scores) we used correlation test Pearson.
Line 226 – Please describe the statistical methods used in the methods section (e.g. information from lines 340-342, 368-380)
Answer: For take in consideration the interactions between 2 variables of sleep and 12 variables of muscles groups, we used statistical analysis MANOVA. This is available in XL STAT software. For compare the means for each variable, we used Wilks Lambda tests(1932), which is the one of the most important test.
Line 235 – “observed” – I suggest using the past tense instead of the present tense.
Answer: DONE
Table 13 and 14 – The values presented in te table are the means for each subgroup, not correlations like the title claims. Please correct it. There are not provided any R and p values to analyze correlations.
Answer: Table 13. Corespondence between each scores group fiecare grup al scorurilor SQS score and the means value of duration of maintained muscle force (knee flexors and knee extensors)
Table 14. Corespondence between each scores group fiecare grup al scorurilor CASQ score and the means value of …average duration of maintain the muscle force (knee flexors and knee extensors)
Line 530-531 – There are still lines from the template. Please correct it.
Answer: DONE
